# Relationship between Physical Activity and Physical and Mental Health Status in Pregnant Women: A Prospective Cohort Study of the Japan Environment and Children’s Study

**DOI:** 10.3390/ijerph182111373

**Published:** 2021-10-29

**Authors:** Yasuyuki Yamada, Takeshi Ebara, Taro Matsuki, Hirohisa Kano, Hazuki Tamada, Sayaka Kato, Hirotaka Sato, Mayumi Sugiura-Ogasawara, Shinji Saitoh, Michihiro Kamijima

**Affiliations:** 1Department of Occupational and Environmental Health, Graduate School of Medical Sciences, Nagoya City University, Mizuho-ku, Nagoya 4678601, Aichi, Japan; yayamada@juntendo.ac.jp (Y.Y.); tmatsuki@med.nagoya-cu.ac.jp (T.M.); h-kano@sass.chukyo-u.ac.jp (H.K.); h-tamada@med.nagoya-cu.ac.jp (H.T.); s-ichiki@med.nagoya-cu.ac.jp (S.K.); h.sato@med.nagoya-cu.ac.jp (H.S.); kamijima@med.nagoya-cu.ac.jp (M.K.); 2Graduate School of Health and Sports Science, Juntendo University, Inzai 2701695, Chiba, Japan; 3School of Health and Sport Sciences, Chukyo University, 101 Tokodachi, Kaizu-cho, Toyota 4700393, Aichi, Japan; 4Department of Pediatrics and Neonatology, Graduate School of Medical Sciences, Nagoya City University, Mizuho-ku, Nagoya 4678601, Aichi, Japan; ss11@med.nagoya-cu.ac.jp; 5Department of Obstetrics and Gynecology, Graduate School of Medical Sciences, Nagoya City University, Mizuho-ku, Nagoya 4678601, Aichi, Japan; og.mym@med.nagoya-cu.ac.jp

**Keywords:** physical activity, pregnant women, prospective cohort study, physical and mental health, JECS

## Abstract

To discuss appropriate physical activity (PA) levels during pregnancy, this prospective cohort study examined the relationships between PA levels before and during pregnancy and physical and mental health status. Fixed data for 104,102 pregnant women were used from the Japan Environment and Children’s Study, of which data for 82,919 women were analyzed after excluding women with multiple birth and pregnancy complications. PA levels were measured using the International Physical Activity Questionnaire-Short Form. The 8-Item Short Form Health Survey was used to measure outcomes. Logistic regression with multiple imputations showed that moderate PA for over 720 min/wk and vigorous PA before pregnancy were associated with poorer mental health in the first trimester (adjusted odds ratio (AOR): 1.087–1.376. Walking in the second and third trimesters was associated with better physical and mental health (AOR: 0.855–0.932). Moderate PA over 1080 min/wk and vigorous PA in the second and third trimesters were associated with poorer mental health (AOR: 1.223–1.873). Increases over 4135.4 MET–min/wk and decreases in PA levels were associated with poorer mental and physical health (AOR: 1.070–1.333). Namely, pregnant women receiving health benefits prefer continuous walking in addition to avoiding vigorous PA and excessive changes in PA levels during pregnancy.

## 1. Introduction

During pregnancy, it is recommended that women have exercise habits and engage in physical activity (PA), as they and their fetuses can receive health benefits, such as the prevention of gestational diabetes [1,2,3], preterm delivery [4], cardiovascular disease [5], and depression [3,6]. Additionally, for pregnant women, exercise leads to improvements in cardiorespiratory fitness, gestational weight management, and sleep cycles, while also preventing urinary incontinence and low back pain [6]. Moreover, performing appropriate levels of PA during pregnancy has not been found to be a risk factor for adverse perinatal outcomes, such as reduced birth weight or increases in preterm birth rates [5,6,7]. Accordingly, evidence-based recommendations for exercise during pregnancy are re-quired [7].

Regarding the recommended PA levels during pregnancy, some researchers have reached a consensus based on strong evidence. For example, the American College of Obstetrics and Gynecology’s (ACOG) recommends that pregnant women engage in PA for at least 20–30 min per day on most or all days of the week [2]. Additionally, the Centers for Disease Control and Prevention and American College of Sports Medicine both recommend that pregnant women participate in 30 min or more of moderate exercise on most, if not all, days of the week [8]. However, especially in the context of findings for keeping sound physical and mental health conditions among pregnant women, due to some research issues there is no consensus among researchers regarding other issues, such as the acceptable range of increase or decline in PA levels during pregnancy, detailed recommended levels of vigorous PA for pregnant women, etc. First, little evidence has been obtained through longitudinal studies assessing the shift of PA levels from before pregnancy to during pregnancy. Accordingly, it is unknown whether there are potential health benefits for either increases or decreases in PA during pregnancy. Second, there is limited evidence regarding the effects of vigorous PA on pregnant women’s health, which has been noted, especially by athletes and sports organizations [9,10,11,12]. Although reviews in the International Olympic Committee report [13] and meta-analysis [14] supported the safety of performing vigorous PA during pregnancy regarding neonatal and pregnancy outcomes, the generalization of research findings and accumulation of evidence regarding pregnant women’s health outcomes are limited. Third, there is still insufficient evidence related to the effects of PA on mental health outcomes specifically for pregnant women [15]. The latest meta-analysis indicated that reduction of PA levels during the COVID-19 epidemic affected an increase in anxiety and depression among pregnant women [16]. However, to date, we do not have specific information on the recommendation range of PA levels for their mental health during pregnant women.

Therefore, to clarify what levels of PA should be recommended during pregnancy, it is necessary to overcome these previous research issues. Therefore, the present study examined the relationships between PA levels before and during pregnancy and physical and mental health during pregnancy using data from a prospective national birth cohort study, the Japan Environment and Children’s Study (JECS) [17,18,19].

## 2. Materials and Methods

### 2.1. Study Design and Setting

The detailed design and baseline characteristics of the JECS have been published elsewhere [17,19]. Briefly, the JECS was a nationwide birth cohort study designed by the JECS Working Group to clarify environmental factors that affect children’s health and development during the fetal period and in early childhood. From January 2011 to March 2014, the JECS recruited approximately 100,000 pregnant women, and then conducted follow-up surveys until the children reached 13 years of age. A total of 15 Regional Centers covering various geographical areas in Japan developed a birth cohort and collected the data.

The JECS was conducted according to the Helsinki Declaration and other nationally valid regulations and guidelines. The JECS protocol was approved by the Institutional Review Board of the Japan National Institute for Environmental Studies (no. 100910001). The JECS protocol was reviewed and approved by the Ministry of the Environment’s Institutional Review Board on Epidemiological Studies and the Ethics Committees of all the participating institutions [17,18,19]. Written informed consent was obtained from all the participants. The present study used the JECS data of the questionnaire research and medical record conducted at two time points, during the participants’ first trimester and within their second and third trimesters.

### 2.2. Participants

The present study used a fixed data set (jecs-ag-20160426) comprising 104,102 fetal records. Pregnant women with multiple birth and pregnancy complications including hypertension, hyperthyroidism, hypothyroidism, diabetes, autoimmune disease, heart disease, kidney disease, hepatitis, cerebral infarction, intracerebral hemorrhage, epilepsy, blood disease, cancer, psychiatric disease, neurological disease, thrombosis, and other pregnancy complications were excluded, since exercise was not recommended for them during pregnancy [2,8]. A flowchart of participant inclusion is shown in Figure 1. A total of 21,239 women were excluded due to pregnancy with multiple births (*n* = 1994), pregnancy complications (*n* = 14,758) or missing data (*n* = 4431). Consequently, valid data from 82,919 women were used for the analysis.

### 2.3. Variables

#### 2.3.1. Exposures

PA levels were measured using the short version of the International Physical Activity Questionnaire (IPAQ) [20,21]. The IPAQ evaluates the frequency and duration of walking, moderate PA, and vigorous PA in the past week [20,21]. PA levels before pregnancy were measured retrospectively in the first trimester, and PA levels during pregnancy were measured in the second or third trimester. On the basis of the official guideline of the IPAQ [22], we conducted data cleaning and created the scores for time spent on walking (min/wk), moderate PA (min/wk), and vigorous PA (min/wk) with a score range of 0–1260 min/wk. These scores were replaced by categorical variables sectioned in increments of 180 min/wk (ref: <10, 10–180, 180–360, 360–540, 540–720, 720–900, 900–1080, 1080–1260). Additionally, we calculated changes in PA levels from before pregnancy to the second and third trimesters (min/wk) using the following equation: Change in PA levels = total metabolic equivalents (METs) min/wk 2nd–3rd trimesters—total MET–min/wk before pregnancy. The numerical variables for changes in PA levels were replaced by categorical variables sectioned in increments of ±0.5 standard deviations (SD) (ref: within ±0.5, 0.5–1.0, 1.0–1.5, 1.5–2.0, ≥2.0, −0.5–−1.0, −1.0–−1.5, −1.5–−2.0, ≤−2.0). Moreover, the procedure of exception of the data for those who have a total days of walking (days/wk), moderate PA (days/wk), and vigorous PA (min/wk) for 8 days or more, proposed by the IPAQ guideline, was not conducted.

#### 2.3.2. Outcomes

The physical and mental health status in the first and second to third trimesters were measured using the Japanese version of the 8–Item Short Form Health Survey (SF8) [23,24,25]. International studies supported the validity and reliability of the SF8 and were used to assess the health-related quality of life [26,27,28,29,30]. The SF8 has eight items and two standardized subscales, the physical health summary score and mental health summary score, which range from 0–100 points (mean score = 50) [23,25]. In this study, standardized physical and mental health scores were replaced by categorical variables. The reference category indicated better health status (score range = 50–100 points) and the case category indicated poorer health status (score range = 0–50 points).

#### 2.3.3. Covariates

The categories for potential confounding variables are shown in Table 1. In a logistic regression analysis, we adjusted for age at pregnancy; body mass index (BMI) during the first and second to third trimesters; annual household income; education level; marital status; experience of stressful events in the past year; experiences of domestic violence during pregnancy; use of fertility treatments; number of previous pregnancies; number of miscarriages, induced abortions or stillbirths; and employment status during the first and second to third trimesters. The variables of BMI, number of previous pregnancies, and number of miscarriages, induced abortions or stillbirths were data of the medical record. Additionally, the other variables were data of the questionnaire research.

### 2.4. Bias

The prospective cohort study design adopted by the JECS had the strength to reduce the effects of recall bias. However, we can assume that there might be recall bias in the evaluation of PA levels before pregnancy, since the deadline for the response of questionnaire was the 21st week of pregnancy. To minimize sampling bias in the national birth cohort study, the JECS selected the study areas. To prevent losses to follow-up, the JECS progressed various public relationship activities to the participants. Furthermore, the study protocol of the JECS was designed and progressed on the basis of original research guidelines to minimize the other study biases. Regarding non-respondent bias, the mean participant age in the missing data was significantly higher than in the valid data. However, the mean difference was very small (t = 20.79, mean diff = 0.89, 95% CI = 0.81–0.97, Cohen’s d = 0.18). Furthermore, mean BMI scores in the valid data of the first trimester and the second and third trimesters were significantly smaller than in the missing data. Similarly, the mean differences were small (the first trimester: t = 16.18, mean diff = 0.50, 95% CI = 0.44–0.56, Cohen’s d = 0.15, the second and third trimesters: t = 16.01, mean diff = 0.51, 95% CI = 0.45–0.58, Cohen’s d = 0.16). Based on these results, the effects of non-respondent bias were comparatively small. Furthermore, this study showed additional results using the multiple imputed data to account for the non-respondent bias.

### 2.5. Statistical Methods

In this study, the binomial logistic regression analysis was used to examine the strength of relationships between exposure and outcome variables. First, the relationships between PA levels before pregnancy and physical and mental health in the first trimester were examined. Second, we examined the associations between PA levels and physical and mental health in the second or third trimester. Finally, the relationships between changes in PA levels after detection of pregnancy and physical and mental health outcomes in the first and second trimesters were examined. The logistic regression was conducted after the missing values were replaced using the multiple imputation method. These analyses were conducted using SPSS version 24.0 (IBM Corp., Armonk, NY, USA).

## 3. Results

### 3.1. Physical Activity Levels before Pregnancy and Physical and Mental Health Status in the First Trimester

First, we examined the relationships between the PA levels before pregnancy and physical and mental health in the first trimester. Table 1 shows the categories of confounding variables adjusted in the logistic regression analysis. Table 2 shows the results of logistic regression. Focusing on adjusted odds ratios (AOR) in multiple imputed data, engaging in walking or moderate PA before pregnancy showed no or a very small relationship with physical and mental health in the first trimester. Performing vigorous PA before pregnancy, especially over 1080 min/wk, was associated with better physical health (vs. inactive: AOR = 0.672, 95% CI = 0.545–0.829). From 360 to less than 1080 min/wk of vigorous PA before pregnancy showed a dose–response relationship with poorer mental health (vs. inactive: AOR = 1.218–1.373).

### 3.2. Physical Activity Levels in the Second and Third Trimesters and Physical and Mental Health Status

We examined the relationships between PA levels and physical and mental health during the second and third trimesters. Table 3 shows the results of logistic regression. As demonstrated by the AORs in multiple imputed data, a dose–response relationship was found between engaging in walking from 180 to 899.9 min/wk and both physical (vs. in-active: AOR = 0.858–0.914) and mental health (vs. inactive: AOR = 0.855–0.932). Neither moderate nor vigorous PA in the second and third trimesters was associated with physical health. However, moderate PA from 180 to 360 min/wk, for over 1080 min/wk and any amount of vigorous PA were associated with poorer mental health (vs. inactive: AOR = 1.094–1.873). Specifically, time spent on vigorous PA in the second to third trimesters showed a dose–response relationship with poorer mental health.

### 3.3. Physical Activity Levels and Physical and Mental Health Status in the Second and Third Trimesters

Finally, we examined the relationships between changes in PA levels during pregnancy and physical and mental health status in the second and third trimesters. Changes in PA levels referred to the gap in total MET–min/wk between the first trimester and the second to third trimesters. In our data, the mean total for MET–min/wk in the first trimester was 1860.178 (SD = 2539.950) and 1130.854 (SD = 1771.269) in the second and third trimesters, and the mean difference between the first and second to third trimesters was 729.323 (SD = 2432.173, 95% CI = 712.489–746.158, t = 84.913, df = 80,185, Δ = −0.29). In this study, the range in the category for maintaining PA was −1945.4 to 486.7 MET–min/wk, on the basis of the mean difference ± 0.5 SD. Table 4 shows the multiple regression analysis results. Compared to the unchanging category, increasing PA levels up to 1702.9 MET–min/wk was associated with higher physical health (vs. unchanging: AOR = 0.906, 95% CI = 0.859–0.955) and mental health (vs. unchanging: AOR = 0.944, 95% CI = 0.899–0.990). However, excessively increased PA levels by over 4135.4 MET–min/wk was related with poorer mental health (vs. unchanging: AOR = 1.313, 95% CI = 1.169–1.473). Moreover, decreased PA levels were related to poorer physical (vs. unchanging: AOR = 1.100–1.162) and mental health (vs. unchanging: AOR = 1.070–1.333). Specifically, decreased PA levels showed a clear dose–response relationship with mental health outcomes.

## 4. Discussion

### 4.1. Appropriate Range of Pre-Pregnancy Physical Activity Levels for Women

In the present study, engaging in walking, moderate PA, and vigorous PA for under 360 min/wk was found to be acceptable for women prior to pregnancy, as these PA levels showed either no or a very small relationship with physical and mental health outcomes in the first trimester. However, engaging in more than 360 min/wk of vigorous PA before pregnancy was related to poor mental health during early pregnancy.

Our suggestions are in line with Tendais et al. [31], who found evidence that women with low PA levels prior to the first trimester of pregnancy showed better mental health outcomes in the first trimester than active women who transitioned to low PA levels in the first trimester. Moreover, suggesting an upper limit of vigorous PA (<360 min/wk) before pregnancy is meaningful, since conventional exercise guidelines prefer to show a lower limit of PA for healthy adult women without assuming the possibility of pregnancy. For instance, the World Health Organization recommends at least 150 min/wk of moderate aerobic PA, at least 75 min/wk of vigorous aerobic PA or an equivalent combination of moderate and vigorous PAs (aerobic PA should be performed in bouts of at least 10 min) for adults aged 18–64 [32]. Additionally, adults should increase their moderate aerobic PA to 300 min/wk or engage in 150 min/wk of vigorous aerobic PA or an equivalent combination of moderate and vigorous PA, to receive additional health benefits [32]. Moreover, The American College of Sports Medicine and American Heart Association have recommended lower limits for adults, suggesting that they engage in moderate-intensity aerobic PA for a minimum of 30 min, 5 days/wk or vigorous-intensity aerobic activity for a minimum of 20 min, 3 days/wk [8]. The Exercise and Physical Activity Reference for Health Promotion and Shibata et al. recommended a total amount of PA with over 23 MET–h/wk (≥1380 MET–min/wk) for non-pregnant adults, as an appropriate lower limit for PA [23,33]. Therefore, based on this study’s results, an appropriate range of PA levels before pregnancy overlapped with the recommended levels for non-pregnant adults in conventional PA guidelines.

### 4.2. Appropriate Range of Physical Activity Levels for Pregnant Women

Pregnant women who gained health benefits walked for longer periods during the second and third trimesters, as this was found to be associated with better physical and mental health during pregnancy. Moderate PA under 1080 min/wk is acceptable, showing either no or a very small relationship with physical and mental health in the second and third trimesters. However, we could not recommend either the time spent on moderate PA with over 1080 min/wk or the longer time spent on vigorous PA in the second and third trimesters, as these levels were associated with poorer mental health.

In the present study, the health benefits of walking and moderate PA during pregnancy were in line with conventional studies and guidelines [9]. For instance, Petrovic et al. [34] confirmed a significant relationship between walking daily and lower risks for depression and anxiety during pregnancy. Moreover, exercise guidelines in Canada, Japan, Norway, Spain, and conventional studies recommend walking during pregnancy to receive general health benefits [9,10]. Regarding moderate PA, the ACOG guidelines for pregnant women recommend setting an eventual goal of engaging in moderate-intensity exercise for at least 20–30 min per day on most or all days of the week [2]. In this case, the recommended time range for moderate PA is 100 min/wk (20 min/day × 5 days) to 210 min/wk (30 min/day × 7 days). Additionally, Canada and the United Kingdom guidelines recommend a minimum of 45 min/wk (15 min × 3 times), progressing to 120 min/wk (30 min × 4 times), even if the intensity was reduced. The Denmark guidelines recommend at least 210 min/wk (30 min × 7 days) of moderate PA [9]. Therefore, the recommendation of an upper limit for moderate PA of 1080 min/wk does not contradict the conventional guidelines [2]. Moreover, providing evidence of a dose–response relationship between vigorous PA and poorer mental health is valuable, since there is little evidence of the discussion of the appropriate upper limit for PA after the detection of pregnancy [11].

Although this kind of evidence is useful, especially for female athletes [12,14], the ac-cumulated evidence from well-designed studies on vigorous PA and pregnancy out-comes has been insufficient [9,10]. Accordingly, conventional guidelines have provided a qualified approval to perform vigorous PA during pregnancy. For instance, the ACOG guidelines show reservations regarding women obtaining approval from their healthcare providers to continue strenuous activity during pregnancy—especially elite athletes who have a clear understanding of the risks—and instead, recommend that women consider decreasing their resistance load compared with the pre-pregnancy conditions [2]. Furthermore, the Canadian guidelines recommend vigorous PA during pregnancy only in a monitored environment [11], while the guidelines in Denmark and the United States recommend that pregnant women only engage in vigorous PA if they regularly did before pregnancy [9].

In addition to healthy active women and athletes, attention should be paid to women with exercise addiction. From the clinical perspective of exercise addiction, performing vigorous PA during pregnancy could be regarded as an expression of addictive behaviors. Generally, people with exercise addiction continue to exercise, regardless of physical injuries, personal inconvenience or disruptions in other areas of life, including marital strain, interference with work, and lack of time for other activities [35]. Their poor mental health can be understood from the perspective of a high risk of depression among people with exercise addiction [35,36]. Therefore, clinicians and other healthcare providers should have specific approaches, especially for healthy active women, for athletes, and for exercise-addicted women during pregnancy.

### 4.3. Appropriate Range for Changes in Physical Activity Levels during Pregnancy

Regarding changes in PA levels during pregnancy, pregnant women who gained physical and mental health benefits in the second and third trimesters tended to maintain or increase PA levels up to 1702.9 MET–min/wk. Moreover, it is acceptable to increase the PA levels up to 4135.3 MET–min/wk. However, excessive increases in PA of over 4135.4 MET–min/wk were found to be associated with poorer health. Furthermore, we discouraged decreasing PA levels during pregnancy under 1945.5 MET–min/wk, in order to maintain good physical and mental health.

The suggestions to maintain or increase PA levels, but avoid excessive increases in PA, support the conventional guidelines and studies suggesting that pregnant women should start new forms of exercise, such as walking, stationary cycling, aerobic exercises, dancing, resistance exercises, and stretching exercises [2,37]. Engaging in appropriate types of exercise, enables pregnant women to stay in shape, maintain their health, keep a routine [13], and prevent antenatal depressive symptoms [38]. Moreover, discouraging excessive increases in PA levels during pregnancy is supported by conventional guidelines. To prevent excessive and rapid increases in PA levels, the Canadian guidelines encourage previously inactive women to increase their PA levels during pregnancy, but caution that they may need to begin gradually, at lower levels of intensity, and then increase the PA duration and intensity as their pregnancy progresses [11]. Additionally, the guidelines in Denmark discourage vigorous PA during pregnancy for women who did not engage in PA prior to pregnancy. Moreover, based on a systematic review, Nascimento et al. [6] suggested that the PA intensity during pregnancy should be mild or moderate for previously sedentary women and moderate to high for active women. In the United States, the guidelines warn that women who are not already highly active or engaged in vigorous PA should focus on moderate PA during pregnancy [9].

For pregnant athletes specifically, the Spanish guidelines have recommended no more than 15 min of vigorous PA, and that the PA intensity should be decreased by 20–30%. Moreover, Pivarnik et al. [39] recommended that pregnant elite athletes should avoid becoming overheated when participating in sports or intense training. Additionally, significant relationships between decreased PA and physical and mental health suggest that exercise during pregnancy is necessary. This problem is especially serious for women who were highly active before pregnancy or athletes who shift from actively training to pregnancy. Generally, elite athletes train at least 5 days/wk, averaging close to 2 h/day throughout the year prior to pregnancy [39]. Furthermore, for recreational and competitive runners, training efforts, intensity, and the number of women who run decreased during pregnancy [13,40]. However, according to our findings, highly active women or athletes should maintain the decreasing PA levels from first to second and third trimesters by less than 1945.1 MET–min/wk, indicating that they should not decline physical activity levels during pregnancy.

### 4.4. Study Strengths and Limitations

A strength of this study is that it provides evidence regarding the upper and lower limits of PA levels for women both before and during pregnancy, in order to receive health benefits during pregnancy. Additionally, this study is valuable in that it shows descriptive and statistical evidence regarding PA and health status among pregnant women using a large sample size. Moreover, the large sample size allowed us to adjust for various confounders in the logistic regression analysis. A prospective cohort study design can minimize recall bias in assessing PA levels and health outcomes. However, a limitation of this study is that our suggestions were based on possible risks to physical and mental health status without other parameters, such as pregnancy complications, pregnancy consequences, and fetus’s health. Furthermore, it is assumed that the health risks could be underestimated, as the present study excluded data for women with pregnancy complications or multiple gestations. Additionally, recall bias regarding the assessment of PA levels before pregnancy using the IPAQ short form during the first trimester of pregnancy is a limitation. However, some previous studies have followed this procedure [4,19,41]. Moreover, the use of the IPAQ short form is one of the best ways to assess a birth cohort study from the perspective of feasibility. This is due to the fact that, in the birth cohort study design, the first trimester of pregnancy is the earliest and most suitable period to measure PA levels in pre-pregnancy. Furthermore, the use of the short form version is appropriate as it does not place undue strain on pregnant women. However, this scale is used to evaluate PA levels within the last 7 days. To overcome this limitation, we recommend future studies to either use objective measuring devices, such as an accelerometry or select other scales that are based on the long-term recall of answers. Although this study selected some confounder variables, it would be preferable for future research to adjust for other potential confounders, such as the change of BMI during pregnancy. Moreover, careful interpretation, in general, is needed in causal inference from observational studies. Since we did not have enough evidence to explain the causal relationship between performing PA and health conditions, further research is warranted to explore the recommended range of PA levels before and during pregnancy based on the new causal inference approach.

## 5. Conclusions

In conclusion, engaging in walking, moderate PA or vigorous PA for under 360 min/wk before pregnancy is acceptable. However, performing vigorous PA for over 360 min/wk before pregnancy is associated with poor mental health during the first trimester. During the second and third trimesters, it might be better to spend more time walking, since moderate PA for over 1080 min/wk or vigorous PA do not lead to better health benefits. Maintaining or increasing PA levels up to 1702.9 MET–min/wk during pregnancy is associated with health benefits in the second and third trimesters. However, excessive increases in PA over 4135.4 MET–min/wk and decreases in PA less than 1945.5 MET–min/wk from the first to the second and third trimesters are associated with poor physical and mental health.

## Figures and Tables

**Figure 1 ijerph-18-11373-f001:**
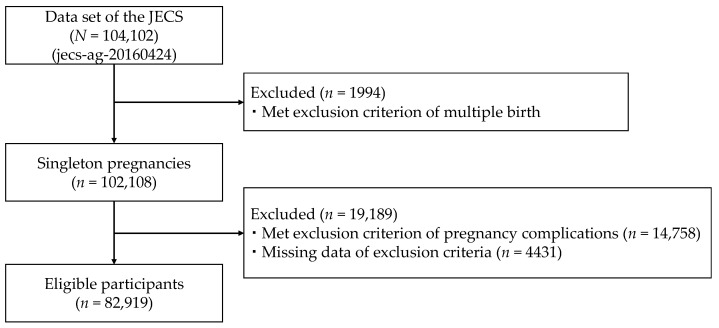
Flowchart of participant inclusion for statistical analysis.

**Table 1 ijerph-18-11373-t001:** Items and categories for potential confounders.

Variables and Categories	*n* = 82,919	
	*n*	(%)
Age at pregnancy		
<20	941	(1.1)
20–29	33,120	(39.9)
30–39	44,607	(53.8)
≥40	2708	(3.3)
Missing	1543	(1.9)
Annual household income (million Japanese yen)		
<2	4249	(5.1)
2–4	26,100	(31.5)
4–6	24,652	(29.7)
6–8	11,742	(14.2)
8–10	4869	(5.9)
≥10	3184	(3.8)
Missing	8123	(9.8)
Mother’s final academic background		
Junior high school	3781	(4.6)
High school	25,637	(30.9)
Higher professional school	1322	(1.6)
Technical college	18,243	(22.0)
Two-year college	13,916	(16.8)
Four-year college	16,102	(19.4)
Graduate school	1145	(1.4)
Missing	2773	(3.3)
Marital status		
Married	77,551	(93.5)
Single	2863	(3.5)
Divorced or Widowed	637	(0.8)
Missing	1868	(2.3)
Causing stressful events of the past year		
No	45,771	(55.2)
Yes	34,271	(41.3)
Missing	2877	(3.5)
Experience of domestic violence during pregnancy		
No	68,934	(83.1)
Yes	11,153	(13.5)
Missing	2832	(3.4)
Undergoing fertility treatment		
No	74,479	(89.8)
Yes	6531	(7.9)
Missing	1909	(2.3)
Number of pregnancies before current pregnancy		
0	24,334	(29.3)
1	28,093	(33.9)
2	17,093	(20.6)
≥3	12,703	(15.3)
Missing	696	(0.8)
Experience of miscarriage or stillbirth before current pregnancy		
No	55,884	(67.4)
Yes	25,671	(31.0)
Missing	1364	(1.6)
BMI ^1^ in the first trimester		
<18.5	13,439	(16.2)
18.5–24.9	60,349	(72.8)
25.0–29.9	6359	(7.7)
30.0–34.9	1399	(1.7)
≥35	342	(0.4)
Missing	1031	(1.2)
BMI in the second to third trimesters		
<18.5	2087	(2.5)
18.5–24.9	61,390	(74.0)
25.0–29.9	13,824	(16.7)
30.0–34.9	2256	(2.7)
≥35	433	(0.5)
Missing	2929	(3.5)
Employment status in the first trimester		
No	29,362	(35.4)
Yes	48,785	(58.8)
Missing	4772	(5.8)
Employment status in the second to third trimesters		
No	36,788	(44.4)
Yes	42,189	(50.9)
Missing	3942	(4.8)

^1^ BMI: Body mass index.

**Table 2 ijerph-18-11373-t002:** Relationship between physical activity before pregnancy and physical and mental health in the first trimester.

PA Levels before Pregnancy	Categories (min/wk)	Poorer Physical Health in the First Trimester						Poorer Mental Health in the First Trimester					
		OR ^1^ (95% CI) ^2^		AOR ^3^ (95% CI)		AOR (MI) ^4^ (95% CI)		OR ^1^ (95% CI) ^2^		AOR ^3^ (95% CI)		AOR (MI) ^4^ (95% CI)	
Time spent walking (3.3 METs)	<10	–		–		–		–		–		–	
	10–180	1.125	(1.080–1.172)	1.079	(1.031–1.129)	1.087	(1.043–1.133)	0.991	(0.953–1.030)	1.033	(0.989–1.079)	1.010	(0.970–1.051)
	180–360	1.078	(1.025–1.134)	1.026	(0.970–1.085)	1.044	(0.992–1.099)	0.988	(0.942–1.037)	1.020	(0.967–1.076)	1.006	(0.957–1.057)
	360–540	0.990	(0.929–1.054)	0.956	(0.892–1.026)	0.969	(0.909–1.033)	0.929	(0.875–0.988)	0.976	(0.912–1.045)	0.934	(0.877–0.994)
	540–720	1.082	(0.986–1.186)	1.064	(0.962–1.177)	1.069	(0.974–1.173)	0.961	(0.880–1.050)	0.960	(0.872–1.058)	0.953	(0.872–1.043)
	720–900	0.997	(0.917–1.083)	1.029	(0.938–1.128)	0.989	(0.910–1.076)	0.966	(0.891–1.048)	0.989	(0.904–1.081)	0.950	(0.875–1.032)
	900–1080	1.030	(0.960–1.106)	1.021	(0.946–1.103)	1.038	(0.967–1.115)	1.079	(1.007–1.157)	1.048	(0.971–1.131)	1.035	(0.964–1.111)
	1080–1260	1.037	(0.971–1.107)	1.042	(0.970–1.119)	1.042	(0.976–1.113)	1.063	(0.997–1.133)	1.040	(0.970–1.116)	1.014	(0.951–1.083)
Time spent on moderate PA ^5^ (4.0 METs)	<10	–		–		–		–		–		–	
	10–180	1.106	(1.059–1.156)	1.083	(1.032–1.136)	1.100	(1.053–1.150)	1.028	(0.987–1.071)	1.029	(0.984–1.077)	1.041	(0.998–1.086)
	180–360	1.059	(0.997–1.125)	1.078	(1.010–1.152)	1.081	(1.018–1.149)	1.055	(0.996–1.117)	1.032	(0.968–1.099)	1.055	(0.995–1.118)
	360–540	1.017	(0.938–1.104)	1.049	(0.958–1.147)	1.051	(0.968–1.141)	1.088	(1.005–1.178)	1.065	(0.975–1.164)	1.071	(0.988–1.161)
	540–720	1.091	(0.983–1.210)	1.147	(1.023–1.285)	1.121	(1.010–1.244)	1.105	(0.999–1.222)	1.089	(0.975–1.216)	1.080	(0.975–1.196)
	720–900	1.021	(0.922–1.131)	1.054	(0.943–1.179)	1.076	(0.970–1.193)	1.175	(1.061–1.301)	1.126	(1.007–1.260)	1.126	(1.015–1.249)
	900–1080	1.014	(0.934–1.101)	1.007	(0.922–1.100)	1.038	(0.955–1.127)	1.183	(1.089–1.285)	1.105	(1.010–1.208)	1.120	(1.030–1.218)
	1080–1260	0.960	(0.869–1.060)	1.008	(0.904–1.124)	1.005	(0.910–1.111)	1.035	(0.938–1.141)	0.938	(0.842–1.046)	0.971	(0.878–1.073)
Time spent on vigorous PA ^6^ (8.0 METs)	<10	–		–		–		–		–		–	
	10–180	1.018	(0.958–1.081)	0.990	(0.927–1.057)	0.988	(0.930–1.050)	1.117	(1.054–1.184)	1.106	(1.038–1.178)	1.087	(1.025–1.153)
	180–360	0.908	(0.830–0.993)	0.880	(0.798–0.970)	0.897	(0.819–0.981)	1.127	(1.031–1.232)	1.082	(0.980–1.193)	1.072	(0.979–1.174)
	360–540	0.992	(0.864–1.139)	0.974	(0.838–1.133)	0.996	(0.867–1.144)	1.284	(1.117–1.475)	1.256	(1.077–1.466)	1.218	(1.057–1.402)
	540–720	0.902	(0.753–1.079)	0.901	(0.739–1.099)	0.914	(0.763–1.095)	1.351	(1.119–1.630)	1.244	(1.011–1.531)	1.247	(1.031–1.509)
	720–900	1.074	(0.882–1.308)	1.192	(0.955–1.486)	1.161	(0.952–1.415)	1.511	(1.231–1.853)	1.410	(1.121–1.774)	1.376	(1.118–1.693)
	900–1080	0.915	(0.803–1.043)	0.948	(0.821–1.094)	0.961	(0.842–1.096)	1.509	(1.310–1.739)	1.364	(1.167–1.594)	1.373	(1.189–1.585)
	1080–1260	0.626	(0.508–0.771)	0.626	(0.496–0.789)	0.672	(0.545–0.829)	1.467	(1.160–1.857)	1.214	(0.934–1.580)	1.239	(0.976–1.574)

^1^ OR: Crude odds ratio. ^2^ 95% CI: 95% Confidence interval. ^3^ AOR: Odds ratio adjusted by age, family income, educational level, marital status, experience of stressful events in the past year, domestic violence during pregnancy, undergoing fertility treatment, number of previous pregnancies, number of miscarriages, induced abortions and stillbirths, BMI in early pregnancy, and work status during early pregnancy. ^4^ AOR (MI): Adjusted OR scored using multiple imputed data. ^5^ Moderate PA: Moderate-intensity physical activity. ^6^ Vigorous PA: Vigorous-intensity physical activity.

**Table 3 ijerph-18-11373-t003:** Relationship between physical activity levels in the second to third trimesters and physical and mental health.

PA Levels during Pregnancy	Categories (min/wk)	Poorer Physical Health in the Second to Third Trimesters						Poorer Mental Health in the Second to Third Trimesters					
		OR ^1^ (95% CI) ^2^		AOR ^3^ (95% CI)		AOR (MI) ^4^ (95% CI)		OR ^1^ (95% CI) ^2^		AOR ^3^ (95% CI)		AOR (MI) ^4^ (95% CI)	
Time spent walking (3.3 METs)	<10	–		–		–		–		–		–	
	10–180	0.968	(0.929–1.008)	0.986	(0.943–1.031)	0.987	(0.947–1.029)	0.919	(0.887–0.952)	0.933	(0.897–0.970)	0.932	(0.899–0.967)
	180–360	0.888	(0.845–0.933)	0.911	(0.863–0.962)	0.912	(0.868–0.959)	0.859	(0.822–0.897)	0.885	(0.843–0.929)	0.866	(0.828–0.906)
	360–540	0.886	(0.831–0.944)	0.920	(0.857–0.987)	0.914	(0.857–0.975)	0.880	(0.831–0.931)	0.881	(0.827–0.938)	0.878	(0.829–0.931)
	540–720	0.840	(0.765–0.922)	0.863	(0.780–0.956)	0.858	(0.782–0.942)	0.905	(0.833–0.983)	0.884	(0.806–0.969)	0.888	(0.816–0.967)
	720–900	0.854	(0.778–0.937)	0.861	(0.778–0.954)	0.866	(0.789–0.952)	0.899	(0.827–0.976)	0.850	(0.775–0.932)	0.855	(0.786–0.932)
	900–1080	0.931	(0.852–1.017)	0.943	(0.855–1.039)	0.917	(0.839–1.003)	0.976	(0.903–1.055)	0.982	(0.901–1.070)	0.964	(0.890–1.045)
	1080–1260	0.881	(0.811–0.957)	0.868	(0.793–0.950)	0.876	(0.806–0.952)	0.958	(0.890–1.032)	0.923	(0.851–1.002)	0.948	(0.879–1.022)
Time spent on moderate PA ^5^ (4.0 METs)	<10	-		-		-		-		-		-	
	10–180	1.003	(0.951–1.058)	0.990	(0.934–1.049)	0.987	(0.935–1.042)	1.017	(0.970–1.065)	0.983	(0.933–1.035)	0.983	(0.937–1.031)
	180–360	0.966	(0.892–1.046)	0.952	(0.873–1.037)	0.961	(0.887–1.041)	1.151	(1.072–1.236)	1.086	(1.004–1.174)	1.094	(1.018–1.177)
	360–540	1.046	(0.937–1.167)	1.074	(0.952–1.211)	1.051	(0.941–1.173)	1.098	(0.997–1.208)	0.997	(0.897–1.109)	1.008	(0.914–1.112)
	540–720	1.142	(0.995–1.312)	1.106	(0.953–1.283)	1.136	(0.988–1.305)	1.182	(1.050–1.332)	1.163	(1.020–1.326)	1.123	(0.994–1.268)
	720–900	1.147	(1.001–1.315)	1.129	(0.974–1.308)	1.137	(0.991–1.305)	1.088	(0.968–1.223)	1.015	(0.892–1.154)	0.990	(0.877–1.116)
	900–1080	1.092	(0.975–1.224)	1.061	(0.938–1.200)	1.053	(0.940–1.181)	1.138	(1.031–1.256)	1.059	(0.950–1.180)	1.069	(0.966–1.183)
	1080–1260	1.008	(0.884–1.149)	1.025	(0.888–1.183)	1.010	(0.885–1.153)	1.372	(1.218–1.544)	1.242	(1.090–1.416)	1.226	(1.085–1.384)
Time spent on vigorous PA ^6^ (8.0 METs)	<10	-		-		-		-		-		-	
	10–180	0.919	(0.809–1.044)	0.914	(0.795–1.051)	0.923	(0.812–1.050)	1.409	(1.254–1.583)	1.250	(1.099–1.423)	1.224	(1.086–1.379)
	180–360	1.073	(0.842–1.368)	1.208	(0.920–1.585)	1.110	(0.870–1.417)	1.469	(1.184–1.822)	1.270	(0.999–1.616)	1.271	(1.019–1.585)
	360–540	1.135	(0.802–1.606)	1.150	(0.785–1.684)	1.138	(0.805–1.610)	1.648	(1.210–2.244)	1.341	(0.951–1.892)	1.350	(0.982–1.856)
	540–720	0.905	(0.636–1.288)	0.939	(0.641–1.375)	0.921	(0.647–1.311)	1.734	(1.242–2.421)	1.497	(1.040–2.153)	1.490	(1.065–2.085)
	720–900	1.173	(0.798–1.725)	1.101	(0.726–1.670)	1.211	(0.823–1.783)	1.881	(1.329–2.664)	1.585	(1.077–2.332)	1.749	(1.226–2.497)
	900–1080	0.959	(0.707–1.302)	0.934	(0.669–1.302)	0.979	(0.720–1.331)	1.737	(1.305–2.312)	1.501	(1.093–2.061)	1.488	(1.110–1.994)
	1080–1260	0.699	(0.422–1.160)	0.732	(0.407–1.315)	0.700	(0.421–1.166)	2.179	(1.268–3.747)	1.550	(0.844–2.847)	1.873	(1.076–3.261)

^1^ OR: Crude odds ratio. ^2^ 95% CI: 95% Confidence interval. ^3^ AOR: Odds ratio adjusted by age, family income, educational level, marital status, experience of stressful events in the past year, domestic violence during pregnancy, undergoing fertility treatment, number of previous pregnancies, number of miscarriages, induced abortions and stillbirths, BMI in the second and third trimesters, and work status in the second and third trimesters. ^4^ AOR (MI): Adjusted OR scored using multiple imputed data. ^5^ Moderate PA: Moderate-intensity physical activity. ^6^ Vigorous PA: Vigorous-intensity physical activity.

**Table 4 ijerph-18-11373-t004:** Relationship between changing physical activity levels during pregnancy and physical and mental health in the second to third trimesters.

Change in PA Levels ^1^	Difference in PA Levels ^2^ (MET-min/wk)	Poorer Physical Health in the Second to Third Trimesters						Poorer Mental Health in the Second to Third Trimesters					
		OR ^3^ (95% CI) ^4^		AOR ^5^ (95% CI)		AOR (MI) ^6^ (95% CI)		OR ^3^ (95% CI) ^4^		AOR ^5^ (95% CI)		AOR (MI) ^6^ (95% CI)	
No Change	±0.5 SD (−1945.5–486.8)	–		–		–		–		–		–	
Increased	0.5–1.0 SD (486.8–1703.0)	0.891	(0.845–0.940)	0.910	(0.859–0.963)	0.906	(0.859–0.955)	0.971	(0.926–1.017)	0.953	(0.905–1.004)	0.944	(0.899–0.990)
	1.0–1.5 SD (1703.0–2919.2)	0.891	(0.815–0.973)	0.882	(0.801–0.972)	0.887	(0.811–0.969)	1.186	(1.095–1.284)	1.047	(0.958–1.143)	1.100	(1.013–1.194)
	1.5–2.0 SD (2919.2–4135.4)	0.996	(0.889–1.117)	1.029	(0.908–1.167)	0.978	(0.871–1.097)	1.157	(1.047–1.278)	1.076	(0.964–1.202)	1.073	(0.969–1.189)
	≥2.0 SD (≥4135.4)	0.901	(0.796–1.019)	0.909	(0.794–1.041)	0.903	(0.797–1.022)	1.503	(1.342–1.684)	1.225	(1.081–1.388)	1.313	(1.169–1.473)
Decreased	−0.5–−1.0 SD (−1945.5–−3161.7)	1.080	(1.014–1.149)	1.119	(1.046–1.198)	1.106	(1.039–1.178)	1.124	(1.065–1.186)	1.076	(1.014–1.141)	1.070	(1.013–1.131)
	−1.0–−1.5 SD (−3161.7–−4377.9)	1.065	(0.987–1.150)	1.092	(1.005–1.188)	1.102	(1.021–1.190)	1.192	(1.116–1.273)	1.125	(1.046–1.211)	1.115	(1.042–1.193)
	−1.5–−2.0 SD (−4377.9–−5594.1)	1.116	(0.997–1.248)	1.246	(1.099–1.412)	1.162	(1.039–1.301)	1.260	(1.145–1.386)	1.128	(1.013–1.255)	1.150	(1.042–1.269)
	≤−2.0 SD (≤−5594.1)	1.010	(0.933–1.094)	1.142	(1.044–1.249)	1.100	(1.015–1.193)	1.553	(1.447–1.668)	1.303	(1.202–1.412)	1.333	(1.239–1.435)

^1^ Change in PA levels: Changing physical activity levels during pregnancy. ^2^ Difference in PA levels: Difference in total MET-min/wk before pregnancy and in the second and third trimesters. ^3^ OR: Crude odds ratio. ^4^ 95% CI: 95% Confidence interval. ^5^ AOR: Odds ratio adjusted by age, family income, educational level, marital status, experience of stressful events in the past year, domestic violence during pregnancy, undergoing fertility treatment, number of previous pregnancies, number of miscarriages, induced abortions and stillbirths, BMI in the second and third trimesters, and work status in the second and third trimesters. ^6^ AOR (MI): Adjusted OR scored using multiple imputed data.

## Data Availability

Data are unsuitable for public deposition due to the ethical restrictions and legal framework of Japan. It is prohibited by the Act on the Protection of Personal Information (Act No. 57 of 30 May 2003, amendment on 9 September 2015) to publicly deposit the data containing personal information. Ethical Guidelines for Medical and Health Research Involving Human Subjects enforced by the Japan Ministry of Education, Culture, Sports, Science and Technology and the Ministry of Health, Labour and Welfare also restrict the open sharing of the epidemiologic data. All of the inquiries on the access to data should be sent to: jecs-en@nies.go.jp. The person responsible for handling the enquiries sent to this e-mail address is Dr Shoji F. Nakayama, JECS Programme Office, National Institute for Environmental Studies.

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
