# Peer review of "Relationship between Physical Activity and Physical and Mental Health Status in Pregnant Women: A Prospective Cohort Study of the Japan Environment and Children’s Study"

_ijerph, 2021, doi:10.3390/ijerph182111373_

Round 1
Reviewer 1 Report
Thank you very much for responding to the revisions.
Author Response
Thank you very much for providing supportive comments and encouragement.
We are thankful for the time and energy you expended.
Reviewer 2 Report
The article allows to discuss the current recommendationsof physical activity in pregnancy. Congratulations.
Author Response
Thank you very much for providing practical comments and encouragement.
We are thankful for the time and energy you expended.
Reviewer 3 Report
I have read your research with interest. Thank you.
It is thought that there must have been many difficulties in conducting a prospective cohort study.
Nevertheless, congratulations on successfully completing this study.
I have made some comments below.
-Introduction section
Physical activity in pregnant women is thought to be related to improved physical function, improved pain, and reduced complications. However, in this study, the relationship between physical activity and mental health status was studied. Please describe the reason in detail in the manuscript.
-Discussion section
In the results of this study, please describe in detail the differences from guidelines in other countries on the appropriate range of physical activity for pregnant women.
Author Response
Response letter to the Reviewer 3
Thank you for the points raised that have helped us improve the quality of our manuscript. Please find below our detailed responses.
Comment 1:
-Introduction section
Physical activity in pregnant women is thought to be related to improved physical function, improved pain, and reduced complications. However, in this study, the relationship between physical activity and mental health status was studied. Please describe the reason in detail in the manuscript.
RESPONSE:
Thank you for your informative comment. Much evidence about the effects of physical activity on mental health outcomes specifically for pregnant women is needed (as stated in Line 65-67, p.2). In accordance with the reviewer’s comment, we inserted the sentence with new reference in the section “1. Introduction” as follows:
“The latest meta-analysis indicated that reduction of PA levels during the COVID-19 epidemic affected an increase in anxiety and depression among pregnant women [16]. However, to date, we do not have specific information on the recommendation range of PA levels for their mental health during pregnant women.”
16) Fan, S.; Guan, J.; Cao, L.; Wang, M.; Zhao, H.; Chen, L.; Yan, L. Psychological effects caused by COVID-19 pandemic on pregnant women: A systematic review with meta-analysis. Asian J Psychiatr 2021, 56, 102533, doi: 10.1016/j.ajp.2020.102533.
Comment 2:
-Discussion section
In the results of this study, please describe in detail the differences from guidelines in other countries on the appropriate range of physical activity for pregnant women.
RESPONSE:
In accordance with the reviewer’s comment, we added an appropriate range of PA stated in other countries’ guidelines in the section “4.2. Appropriate Range of Physical Activity Levels for Pregnant Women” as follows:
“Exercise guidelines in Canada, Japan, Norway, Spain and conventional studies also recommend walking during pregnancy to receive general health benefits [9,10].”
“Additionally, Canada and the United Kingdom guidelines recommend a minimum of 45 min/wk (15 min × 3 times), progressing to 120 min/wk (30 min × 4 times) even if intensity was reduced. Denmark guideline recommends at least 210 min/wk (30 min × 7 days ) of moderate PA [9].”
This manuscript is a resubmission of an earlier submission. The following is a list of the peer review reports and author responses from that submission.
Round 1
Reviewer 1 Report
This study sought to examine the associations between varying levels of physical activity before and during pregnancy with physical and mental health during pregnancy. I commend the authors for undertaking this study. I have some concerns regarding the conclusions the authors have drawn from the data, and with the methods for determining physical activity prior to pregnancy. Detailed feedback on the manuscript is provided below.
INTRODUCTION
Line 54: Please provide examples of "other issues related to PA and pregnancy"
Lines 57-58: There is evidence of improvements of cardiovascular function (such as VO2 max) in pregnant women engaging in moderate to vigorous activity.
Line 60: The IOC series on exercise and pregnancy has reported that exercise at less than 90% of VO2max during pregnancy has been shown to be safe. While I agree that evidence is limited, there is some available evidence and it should be cited.
MATERIALS AND METHODS
Lines 102-103: How far into the first trimester were participants asked to recall pre-pregnancy physical activity? The IPAQ is not validated for reporting physical activity beyond the previous 7 days. Use of the IPAQ to estimate pre-pregnancy PA is a methodological flaw and should be mentioned in the limitations section of the paper. There are questionnaires designed to quantify physical activity over the previous 12 months, which would have been a more valid choice for quantifying pre-pregnancy PA.
Line 139: the authors state that the sample size alone is enough to overcome recall bias, however, the use of an inappropriate outcome measure for pre-pregnancy PA introduces significant risk of recall bias.
RESULTS
Lines 182-184: "Neither moderate PA nor vigorous PA in the second and third trimesters was not associated with physical health." This sentence does not make sense.
Table 1: What does "engaging work" mean?
DISCUSSION
Line 222-223: The cited research is not saying that an upper limit of PA should be set prior to pregnancy; instead, it is suggesting that active women should maintain their activity levels during the 1st trimester of pregnancy!
I have significant concerns with the authors suggestions throughout the discussion that the levels of physical activity are the CAUSE of the poor mental health of the participants. It is equally possible that high levels of PA are a SYMPTOM of poor mental health, as the authors highlight in lines 273-282. Suggesting an upper limit of PA for non-pregnant and pregnant women without significant evidence of a CAUSAL relationship is dangerous and irresponsible. It would be appropriate to highlight the association, but not to suggest that PA is the cause. It would also be prudent to highlight in the discussion the overwhelming need to address mental health in pregnant women directly, as they may be using physical activity as an attempt to decrease stress/depression/anxiety, etc.
Reviewer 2 Report
Dear Authors.
It is my honor to review your manuscript for this issue. It is worth imagining that it was very difficult to recruit subjects since a sufficient sample size is considered.
Here are some of the points that caught my attention in the valuable research report presented by you.
1.(p1) " During pregnancy, it is recommended that women have exercise habits and engage in physical activity (PA), as they and their fetuses can receive health benefits, such as the prevention of gestational diabetes [1−3], preterm delivery [4], cardiovascular disease [5], and depression [3,6]. Additionally, for pregnant women, exercise leads to improvements in cardiorespiratory fitness, gestational weight management, and sleep cycles, while also preventing urinary incontinence and low back pain [6]. Moreover, performing appropriate levels of PA during pregnancy has not been found to be a risk factor for adverse perinatal outcomes, such as reduced birth weight or increases in preterm birth rates [5−7].
Accordingly, evidence‐based recommendations for exercise during pregnancy are required [8]. "
A part of "Moreover, performing appropriate levels of PA during pregnancy has not been found to be a risk factor for adverse perinatal outcomes, such as reduced birth weight or increases in preterm birth rates [5−7].", given the connection between the preceding and following sentences, it is unclear what this sentence is trying to say. Are you saying that while exercise for pregnant women can improve cardiorespiratory function, weight control during pregnancy, improve sleep cycles, and prevent urinary incontinence and back pain, there are previous studies that show that an appropriate level of exercise during pregnancy is rather poor?
2(p2). Approximately 20% (n = 19,189) of the population with singleton pregnancies (n = 102,108) were excluded from the analysis. Is there a possibility of bias due to missing data or other exclusions? Please consider conducting the multiple imputation.
3(p3).Are there any differences in the results depending on whether or not these women have given birth before?
4(p3). Regarding the mental health score, is it affected by the time of year (seasonality) of the test performed?
5 (p5).Of those analyzed, more than 50% were in their 30s. On the other hand, although the percentage is small, there are people under 20 years old and people over 40 years old. I understand that you are using age as a covariate in your analysis, but is there any difference in the amount of physical activity according to age in the first place?
6. Have you considered the differences in physical activity depending on the type of residence?
Reviewer 3 Report
The article provides a relationship between two measures using two surveys in a sample of a large number of pregnant women with different characteristics. It is recommended to discuss a little more the future analysis of subpopulations of pregnant women according to increase in BMI and their level of physical activity and what other studies have found using cohorts with accelerometry.Author Response
See the attached file.

Round 2
Reviewer 1 Report
I appreciate the authors' responses to reviewer feedback. However, I continue to have serious concerns about the methodology of the study and the conclusions drawn. As previously mentioned, I do not feel that the use of the IPAQ to quantify pre-pregnancy PA (especially at 21 weeks of pregnancy) is appropriate, and presents a major confounder to the data. Second, I have significant concerns about grouping the data into categorical variables rather than just comparing the raw scores. In addition, the SF-8 reports the patient's perception of physical health, so individuals with significant anxiety or depression may perceive their physical health as worse than it actually is, but no data is presented from the medical record to support whether or not their physical health truly is worse.